# A Summary of Practical Considerations for the Application of the Steric Exclusion Chromatography for the Purification of the Orf Viral Vector

**DOI:** 10.3390/membranes12111070

**Published:** 2022-10-29

**Authors:** Friederike Eilts, Keven Lothert, Sabri Orbay, Felix Pagallies, Ralf Amann, Michael W. Wolff

**Affiliations:** 1Institute of Bioprocess Engineering and Pharmaceutical Technology, University of Applied Sciences Mittelhessen (THM), Wiesenstr.14, 35390 Giessen, Germany; 2Department of Immunology, University of Tuebingen, Auf der Morgenstelle 15, 72076 Tuebingen, Germany; 3PRiME Vector Technologies, Herrenberger Straße 24, 72070 Tuebingen, Germany; 4Fraunhofer Institute for Molecular Biology and Applied Ecology (IME), Ohlebergsweg 12, 35392 Giessen, Germany

**Keywords:** hofmeister series, membrane chromatography, polyethylene glycol precipitation, pore size, regenerated cellulose, steric exclusion

## Abstract

Steric exclusion chromatography (SXC) is a promising purification method for biological macromolecules such as the Orf virus (ORFV) vector. The method’s principle is closely related to conventional polyethylene glycol (PEG) precipitation, repeatedly implementing membranes as porous chromatographic media. In the past decade, several purification tasks with SXC showed exceptionally high yields and a high impurity removal. However, the effect of varying process parameters, on the precipitation success and its limitations to SXC, is not yet well understood. For this reason, the precipitation behavior and SXC adaptation for ORFV were investigated for the PEG/ORFV contact time, the membranes pore size, and the type and concentration of ions. All three parameters influenced the ORFV recoveries significantly. A small pore size and a long contact time induced filtration effects and inhibited a full virus recovery. The application of salts had complex concentration-dependent effects on precipitation and SXC yields, and ranged from a complete prevention of precipitation in the presence of kosmotropic substances to increased efficiencies with Mg^2+^ ions. The latter finding might be useful to reduce PEG concentrations while maintaining high yields. With this knowledge, we hope to clarify several limitations of SXC operations and improve the tool-set for a successful process adaptation.

## 1. Introduction

The application of biological nanoparticles, such as viruses, is increasingly in the focus of the biopharmaceutical industry. Compared to proteins as active pharmaceutical ingredients, new challenges have been encountered in the production processes of viruses, especially in downstream processing. This is mainly due to the different composition and size of virions. Thus, the unit operations in downstream processing need reconsideration. For chromatographic purification tasks, convective mass transport, e.g., in membranes or monoliths, is often preferred over resin-based applications, offering primarily diffusion [1,2]. A rather new chromatographic method, which frequently implements membranes for virus purification, is the steric exclusion chromatography (SXC). The SXC is closely related to precipitation [3], which is intrinsically described by the theories of attractive depletion and excluded volumes [4]. Precipitation is strongly dependent on self-association effects [5], which alter the interaction of the solvents (including the targets) among each other [6]. Additionally, precipitation is adaptable to nearly all biological nanoparticles and large proteins, by varying the process parameters, e.g., solvent concentrations, incubation time, or ionic strength [7]. However, the impurity removal in precipitation is often insufficient and a solid-liquid separation needs to be performed subsequently [8]. For this reason, chromatographic processes are preferred over precipitation in modern pharmaceutical production, due to their selectivity and scalable throughput [3]. Steric exclusion chromatography (SXC) combines the approaches of precipitation and chromatography and several advantages are offered: the flow rates and, therefore, throughput are adjustable, continuous processing is possible, and a secondary step for the separation of the precipitates from the supernatant is omitted [3,8]. SXC was first described by Lee et al. [9]. Since then, applications with various biological macromolecules and nanoparticles, such as immunoglobulins [8,9,10,11], viruses [9,12,13,14,15,16,17,18,19,20], virus-like particles [21], latex particles [22], and nucleic acids [23] were published. Excellent introductions into the working principles of SXC were published by Lee et al. [9] and Gagnon et al. [10], which we encourage to consult. 

As the name suggests, SXC is based on exclusion effects. In most applications, polyethylene glycol (PEG) is utilized as the precipitating agent. Just as in conventional PEG precipitation, by using PEG of an appropriate size, i.e., hydrodynamic radius [7,24], and concentration, the target molecules are excluded from the bulk solution according to their respective size [4,5]. Conclusively, the effect is enhanced by increasing the size and the concentration of the solutes, i.e., PEG and the target [5]. As the PEG molecules are unable to fill the space of preferential hydration around each target, an increase of water concentration in this region is caused compared to the bulk zone, so that a so-called PEG-deficient zone is formed [9]. By this discontinuity, the free energy of the system is increased, and the unfavorable status is resolved by a reduction of the accessible interface, which forces the targets to associate. Up to this point, the SXC process is no different from conventional precipitation. However, in the chromatographic process, a stationary phase, e.g., a membrane or monolith, is implemented. In other words, the stationary phase is part of the precipitating system, by offering a large surface area, where a PEG-deficient zone can form. Hence, the targets form precipitates at the inner surface area of the stationary phase and are retained [9], making the SXC a bind-and-elute method. Concerning the workflow, the target is combined with a PEG-enriched solution and loaded onto a chromatographic column, e.g., a hydrophilic membrane (Figure 1). Whereas conventional PEG-precipitation aims to maximize selective agglomeration, SXC is based on the retention of the target on the stationary phase by the same exclusion effects and contaminants are washed out throughout the loading of the target. By an omission of PEG in SXC, the target can be eluted from the column with a minimized residual PEG concentration in a purified concentrated manner. At this point, it should be made clear that SXC is not based on any kind of filtration process. In fact, filtration is a limitation to the method. Thus, the balance among rapid precipitation, sufficient retention of the target, and pore blockage by big precipitates must be maintained. Other limitations are an increased viscosity of the PEG-solutions, which can cause reduced diffusion and an increased pressure in the chromatographic system. Apart from the PEG and the target size and concentration, other important factors for the here presented SXC application are the surface characteristics of the targets and the stationary phase (charge [5,9], hydrophilicity [24]), composition of the liquid phase (buffers and salts [5,24,25,26]), and the contact time of target and PEG [18]. 

One target that was successfully purified by SXC is the Orf virus (ORFV). The ORFV is a DNA-virus of the genus *Parapoxvirus* that has an ovoid shape of approximately 140–200 nm in width and 220–300 nm in length [27,28,29]. The pharmaceutical application of the ORFV was shown as a viral vector platform [30,31,32,33], an oncolytic and antiviral therapy [34,35,36], and as an immunomodulatory agent [37,38,39]. Like many other poxviruses, the ORFV is very stable against degrading forces, and thus an ideal model for purification processes. This work is based on two recent publications, where we reported on the application of SXC in a purification process for ORFV, implemented in a full downstream processing (DSP) train [13,14]. In this process, SXC was employed with yields of >80% and a full recovery from the column as well as a high impurity removal. Here, we describe the process of a SXC adaptation for the purification of infectious ORFV with cellulose membrane columns, in consideration of these published results. We investigated three parameters: the PEG/ORFV-contact time, the pore size of the membrane stationary phase, as well as the composition and concentration of different salts and buffers. Complimentary to the SXC application of the ORFV, offline aggregation kinetics of the SXC load were performed and analyzed. These data were used to understand the column blockage events and the reduction of yields due to a reduced target retention. We want to give an insight into the adaptation of SXC for a purification task, and equip the reader with pertinent tools to adjust the method for possible new targets with membrane columns. We further discuss the possibility to use precipitation kinetics to predict optimal parameters for a SXC application. With this knowledge, we hope to pave the path for SXC as a possible future purification platform technology.

## 2. Materials and Methods

All chemicals were purchased from Carl Roth, if not stated otherwise.

### 2.1. Preparation of Virus Stocks

In this study, the attenuated ORFV strain D1701-V, expressing the green fluorescent protein AcGFP [31], was used. The virus was produced in Vero cells, as previously described [32]. After a successful cell infection and incubation, the cell-culture was subjected to one freeze-thaw cycle (−80 °C). A clarification from cell debris was undertaken, using centrifugation (4500× *g*, 30 min). Subsequently, the supernatant was stored at −80 °C until further use. For each set of experiments, several batches from this procedure were pooled to generate one stock. 

### 2.2. SXC Application

The SXC of the ORFV was performed as described several times before [12,15,16,17] with minor adjustments. In brief, the stationary phase was comprised of single-use regenerated cellulose membranes, combined in a stack of ten sheets with 13 mm in diameter. The nominal pore size was 1 µm (Whatman, Maidstone, UK), and alternatively 3–5 µm (kindly provided by Sartorius Stedim). The column was equilibrated with a run buffer (either 20 mM TRIS-HCl, 0.1 m HEPES, PBS, or 0.1 M citrate phosphate buffer (CPB)), combined with the desired salt (0–600 mM of Na_2_SO_4_, KCl, NaCl, NaNO_3_, MgSO_4_, or MgCl_2_) and the appropriate PEG concentration (2–8%) and PEG molecular weight (6000–12,000 Da). For loading, the clarified cell culture supernatant was mixed 1:4 with a concentrated buffer to match the concentrations of the run buffer. To generate the feed, offline or in-line mixing was applied. In detail, for offline mixing, the ORFV/PEG solution was prepared in advance and filled into a superloop (Cytiva, Marlborough, MA, USA) and loaded onto the column. For in-line mixing, the ORFV and the enriched PEG solutions were pumped simultaneously and combined in the mixer (1.4 mL) of the FPLC system (Äkta Pure 25, Cytiva). If not stated otherwise, the feed was loaded onto the membranes until the pressure limit of the respective membrane module was reached, i.e., 2.0 MPa for 1 µm membranes modules and 0.4 MPa for 3–5 µm modules. After loading the sample onto the column, a wash step with the run buffer was undertaken for at least 5 mL. Subsequently, the elution step was performed by omitting the PEG and the additional salts, however, 0.4 M NaCl was added to the buffer. All buffers used in the SXC experiments were of neutral pH (7.4) and their influence on the ORFV infectivity was neglectable for the duration of the SXC runs. A flow rate of 2 mL min^-1^ was used for all runs, unless stated otherwise. Relative recoveries of the infectious ORFV were determined, based on the concentration of each initial load sample.

### 2.3. ORFV Infectivity Titration

The infectivity of the ORFV samples was determined by a flow cytometry assay using a Guava easycyte HT (Merck Millipore, Darmstadt, Germany). The procedure was adapted from Lothert et al. [14], however, the washing procedure was omitted. Instead, after centrifugation, the cell pellets were resuspended with PBS, containing 1% paraformaldehyde, 2% FCS, and 2 mM EDTA (VWR International, Radnor, PA, USA). The readout was conducted within 48 h. The relative error between samples in one batch was below 10%. The effect of PEG on the ORFV infectivity was assumed to be neglectable, proven by full recoveries of infectious ORFV in previous experiments [13,14]. The presence of all tested buffers and salts (Na_2_SO_4_, KCl, NaCl, NaNO_3_, MgSO_4_, or MgCl_2_) applied in the SXC experiments had no effect on the virus infectivity (Appendix A) or the assay (Appendix A).

### 2.4. Size Measurements

The precipitation behavior of the clarified ORFV solutions was analyzed over time. The cell culture supernatant was mixed with a concentrated PEG/salt-solution in a relation of 1:4 to equal the desired concentrations. Two different buffering systems were used for these experiments: 20 mM TRIS-HCl (pH 7.4) was applied for all experiments with varying salt concentrations (20–200 mM), and 0.1 M CPB of physiological conductivity, on the other hand, was used for screening different PEG concentrations (0–12%) and pH values (4–7.4). Immediately after preparation, the samples were analyzed by dynamic light scattering (DLS), using a Zetasizer ZS90 with the respective Zetasizer software, version 7.13, (both Malvern Panalytical) in micro-cuvettes (Sarstedt, Nümbrecht, Germany). Automatic size measurements were conducted every 5 min over the course of up to 14 h. After the measurement, the precipitates were visualized by bright-field microscopy (MC170HD with Leica Application Suite software, version 3.0.0, both Leica). Next to this data acquisition, a design of experiments (DOE)-based model was generated for the time-dependent size change of the samples under varying pH and PEG_6000_ (molecular weight of 6000 Da) concentrations (see Section 2.5). 

### 2.5. Statistical Analysis

Unless stated otherwise, the experiments were carried out in triplicates (*n* = 3), which were used to calculate the respective mean and standard deviation. For an analysis of significance, an ANOVA with *Tukey* test was performed, where appropriate (Origin Pro 2021b, OriginLab Corporation).

Furthermore, two DOE models were generated with the Design Expert software (version 12, Stat-Ease). These included screening for critical process parameters of the SXC (Section 3.1.1, PEG concentration, and flow rate), as well as a depiction of the aggregation kinetics of the ORFV in the presence of PEG and varied pH values (Section 3.1.3, time). The statistical evaluation of these designs can be found in the Appendix A.

## 3. Results and Discussion

### 3.1. Influence of the SXC Loading Parameters on the ORFV Recoveries

In this chapter, the loading parameters, i.e., PEG concentration and molecular weight, as well as the flow rate of the SXC experiments with the ORFV, were evaluated. Additionally, the extent of time-dependent precipitation, analyzed by offline size measurements, was compared with the ORFV recoveries and yields in the SXC.

#### 3.1.1. PEG Concentration and Flow Rate

First, using a DOE-approach, different PEG molecular weights were tested in the SXC of the ORFV, according to previously applied concentrations and molecular weights for virus purification [16], i.e., 4–8% PEG of 6000 to 12,000 kDa. For each run, 10 mL ORFV/PEG sample was loaded onto the membrane modules. All experiments were done using a TRIS buffer with 180 mM NaCl. The virus was mixed offline, and the time of mixing, until the entire sample was loaded onto the column, was approximately 20 min. A linear relation was observed, as increased yields were obtained with an increasing PEG concentration and molecular weight from 41% up to 64%, with an optimum at the edge of the design space at 8% PEG_12000_ (Figure 2A). In other words, as the maximum recovery was obtained with the highest chosen PEG concentration and molecular weight, it could be possible that the optimum lies outside the tested range of these parameters. The relation was dominated by the PEG concentration, indicated by a steeper slope for an increasing yield, i.e., a 15% yield increase between 4% and 8% PEG versus a 5% yield increase between 6000 and 12,000 kDa PEG. This was expected, as the precipitation of a target increases in efficiency related to higher PEG concentrations [5] up to a certain point, where precipitation is limited by a compression of the PEG molecules. This causes smaller hydrodynamic radii fo the PEG molecules and reduces the efficiency of exclusion phenomena [7]. In this research work, the maximum applicable PEG concentration for the system was not found. The optimum for an ORFV yield was not covered by the tested design space, presumably lying at higher PEG concentrations and molecular weights. Higher PEG concentrations were not tested, as a high viscosity hampers processing and prevents the application of high flow rates during chromatography. 

Overall, the ORFV recoveries in the qabove descibres elution fraction were lower than published before for the same column type of regenerated cellulose membranes with 1 µm pore in diameter [14], i.e., >80% using 8% PEG_6000_. We assume this to be due to an increased flow rate (Appendix A) and a fixed loading volume of ORFV, which corresponded to only half the dynamic binding capacity of the membrane modules determined with 1.81 × 10^8^ IU (infectious unit) mL^-1^_membrane_ by Lothert et al. [14]. 

It should be noted that no extensive pressure increase throughout the sample loading, apart from viscosity-induced increases, was observed with any tested PEG concentration. This led to two conclusions. Firstly, the precipitation in the pore system of the stationary phase, which is the working principle of the SXC, did not reach a maximum to cause a pore blockage. This was not surprising, as the binding capacity of the column was not fully exploited. Secondly, precipitates, which formed throughout the holding time of sample loading, caused no adverse pressure effects. Using offline mixing, as for the experiments in this section, the loading time of the pre-mixed ORFV/PEG solution onto the column took 20 min. Hereafter, this time interval is called *incubation time*. This period is expected to allow for the formation of ORFV precipitates, which increase in size over time while the load is applied. From this description, one might apprehend pore blockage of the stationary phase due to aggregates, which are held back on the upper layers of the membranes. With the here applied set-up, no such filtration effects were indicated by the online pressure monitoring, however, higher loading volumes might reveal the opposite.

#### 3.1.2. Incubation Time Using PEG_6000_

To further investigate the impact of the incubation time on the ORFV precipitation kinetics, the time-dependent size distribution of the cell culture-derived ORFV, combined with PEG, was measured in static experiments by DLS. We tested 0–12% PEG_6000_ in CPB, depicted in Figure 2B,C for pH 7.4. Other pH values tested in the DOE-planned size measurements are presented in the Appendix A, especially Appendix A.

For all tested PEG concentrations, the aggregate size increased with time. However, the slope, i.e., the velocity of the aggregate size increase, decreased to approximately zero, approaching 60 min for all tested PEG concentrations (Figure 2C). As for the mean sizes after 60 min incubation time, the DOE model predicted the strongest PEG-induced impact between 3% and 8% PEG_6000_ (Figure 2B). The visualization of the ORFV aggregates by bright-field microscopy revealed the same trends as predicted by the statistical model (see Appendix A). The dimensions of the aggregates were of random shape, seemingly consisting of clusters of smaller circular particles (5–10 µm). This leads to the assumption that precipitates form around nuclei, such as cell debris, and grow and randomly attach to each other [40]. The accumulations of smaller precipitates in loose formations might be more sensitive to a shear stress-induced break-up, rather than big dense aggregates [41]. Accordingly, we assume that the actual size of the aggregates loaded onto the membranes is smaller than the size measured in the static experiments shown here. Thus, this effect might be the cause for the precipitates of virus and cell debris to not increase the pre-column pressures significantly by inducing pore blockage in the top membrane layers. 

Surely, this set-up is limited by the DLS method itself, as the size distribution often does not resolute different size fractions, especially for aggregates exceeding 6 µm [10]. Methods, investigating the aggregate density and formation, could improve the validity of the aggregation kinetics [40]. However, from these experiments we expect to derive an indicator for the precipitate formation, depending on the changes in the solute composition.

#### 3.1.3. Incubation Time Using PEG_8000_

Using the knowledge from the previous section, two extrema for the incubation time of an ORFV/PEG_8000_ mixture in CPB were tested in the SXC experiments: No incubation, and 12 h incubation time, both using offline mixing. To facilitate the analysis, the load volume for the column was set again to 10 mL feed. Without incubation, the total ORFV recovery was 102 ± 30%, whereas an incubation time of 12 h revealed 66 ± 20% (Figure 2D). The recoveries were calculated according to the respective load sample. Thus, a mean of 34% was not recovered from the column for the runs after 12 h incubation time. The ORFV yield in the elution fraction with no incubation time showed 67 ± 20%, whereas 12 h incubation time showed 42 ± 8%. The observations were supported by online DLS monitoring, which increased by 100 mV during the loading phase for both incubation times (Appendix A). No differences were observed in the online pressure measurement, similar to the results in the previous section. Thus, no elevated pore blockage is suspected due to increased incubation durations and increased precipitate size. It should be noted that the molecular weight of the PEG was 8000 Da in this set-up (Figure 2D,E), whereas 6000 Da were used for the experiments in Figure 2(A–C1). The increase in molecular weight entailed an increase of the mean size of the precipitates between a mean of 2500 µm up to nearly 4000 µm after 60 min (compare Figure 2C,E). 

Nevertheless, the reduction in total recovery and yield indicated remaining virions on the membranes. Adsorption to the cellulose filters was excluded, as they are known to poorly bind viruses [42] and such effects were not observed in previous experiments with the ORFV [13,14]. Therefore, we assume that the increased precipitate sizes after a prolonged incubation time induced a retention of the ORFV on the upper layers of the membrane stationary phase. In other words, the ORFV precipitates were filtered instead of being retained by exclusion effects. A similar, but more pronounced reduction in the total recovery, due to the incubation of the virus/PEG mixture prior to column loading, was reported by Labisch et al. [18] for a lentivirus vector. The authors suspected the difference to originate from the formation of precipitates prior to loading onto the column, which we confirm here for the ORFV vector. Although our previous works with the ORFV showed no limitations using offline mixing [13,14], and the total virus recovery with offline mixing (0 h incubation time) revealed up to 100% total ORFV recovery, the in-line mixing approach, as recommended by the lentivirus report and other publications [9,17,18], remains to be evaluated for the ORFV.

### 3.2. Variation of the Pore Size of the Stationary Phase

Based on the results in the previous section, which indicated possible filtration effects for ORFV precipitates in offline mixing, an in-line-mixing set-up was tested for two pore sizes of the cellulose membrane columns, i.e., 1 µm, and 3–5 µm. The mixing of the enriched PEG solution and the virus solution was done in the tubing of the chromatographic system, directly before loading onto the column. According to previous experiments in Section 3.1.1, the buffer was a TRIS buffer with 180 mM NaCl and 8% PEG_6000_. Surprisingly, the loading of the sample onto the 1 µm cellulose membranes was not possible with a constant backpressure, and the loading volumes varied between 3 and 16 mL, using the same virus stock. However, the yields remained high with 99% ± 17% (Table 1). By changing the pore size from 1 µm to 3–5 µm, the loading volume was increased by more than three-times (Table 1). The virus load that was quantified in the flow-though, however, increased by 30%. Overall, the 3–5 µm membranes showed a robust operation with small standard deviations. 

Explanations for these observations for the 1 µm membranes could be sought in the nature of the chromatographic column, which was single-use and assembled manually. Thus, variations in the individual set-up can be expected. Additionally, the 3–5 µm membranes were of reinforced cellulose, adding more stability to the column bed. We assumed that by using in-line mixing, the PEG/ORFV suspension was not completely homogenous, leading to spontaneous increases in pressure due to higher local viscosities. The bigger pores with a lower backpressure might have been less affected by this behavior and allowed for higher loading volumes [11]. Last, the increased virus concentration in the flow-through, and thus reduced yields, could be due to a limited spontaneous encounter of the precipitating targets and the stationary phase inside the matrix with bigger pores, leading to a reduced accretion of the virions. 

Another interesting observation throughout this experimental set-up was the recovery in the two elution fractions, i.e., down-flow (in the flow direction of loading) and up-flow (counter direction), which were performed consecutively, as previously described [15]. A considerable fraction of the ORFV was eluted from the 1 µm membranes (20%), whereas the yield from bigger pores increased by only 5%. This leads to the assumption that for the 1 µm membranes, ORFV accumulated in or on the upper layers of the membranes, leading to a limitation similar to an increased incubation time prior to loading (see Section 3.1.3). Presumably, the big size of the Orf virions (up to 200 × 300 nm [27,28,29]) caused this effect, which has not been reported before for other viruses using in-line mixing, such as for the bacteriophage M13K07 (7 × 900 nm) [9], the lentivirus (80–100 nm) [18], or the adeno-associated virus (25 nm) [17]. Hence, pore sizes of the stationary phase need a careful validation to prevent filtration effects and to maximize loading capacities in SXC. We further concluded that in-line mixing is a valid alternative to the previously applied offline mixing for the ORFV [13,14]. 

### 3.3. Comparison of Buffering Systems in the SXC Process

Throughout our experiments, using CPB and TRIS buffering systems, a tendency for reduced yields using CPB instead of TRIS was observed for different batches of the ORFV. We systematically investigated the SXC performance with different buffers, i.e., CPB, TRIS, PBS, and HEPES (Figure 3). All buffers were of neutral pH, and supplemented with NaCl for a physiological conductivity of 15 mS cm^-1^. To facilitate a comparison with the 1 µm membrane modules, the load volume was set again to 10 mL. The highest yields were achieved in the presence of PBS (91 ± 19%) and HEPES (82 ± 10%). The application of CPB showed a significantly reduced yield as compared to PBS (Δ 26%) and HEPES (Δ 17%), as well as increased concentrations in flow-through and wash. The online DLS analytics support these trends with according increases (Appendix A). By fractionizing the elution, the major share of the ORFV was detected in the first 4 mL of elution for all buffers (Figure 3). This curtailment allowed for a concentration factor of 2.5× compared to the applied volume of 10 mL The additional elution of 6 mL increased the recovery of infectious ORFV by only 5%. Thus, the total concentration factor with a 4 mL elution volume was 1.9×, considering the 0.25-times dilution of the virus solution to generate the PEG-enriched feed.

Next to the chromatographic experiments, offline precipitation kinetics were conducted. A similar approach was described before by Gagnon et al. [43] for hydrophobic interaction chromatography. The authors pointed towards the positive correlation of precipitation efficiency and retention on such stationary phases. In our study, the differences among the buffers, concerning the ORFV yield, were reflected by offline precipitation kinetics. A slightly smaller size (Δ 500 nm) of the precipitates, incubated in CPB, was measured, as compared to TRIS (data not shown). This observation should be attributable to the addition of the citrate component as well as the higher concentration (two-times) of phosphate, as PBS showed no analogous deviations. Both, citrate and phosphate, are strongly kosmotropic salts. It was reported before that kosmotropic substances induce a stronger effect in inhibiting the target retention throughout SXC [9]. The complex nature of solute interactions and the influence of such additives is further discussed in Section 3.4. It can be concluded here that the choice of the buffering system has a significant influence on the SXC performance, and should be made with care.

### 3.4. Changes in the ORFV Yield Depending on Salt Addition in the SXC Load

After the evaluation of different buffering systems, an ion-specific effect on the ORFV yield in the SXC and the precipitation success was expected. To explore this further, we assessed salts with varying concentrations according to their position in the Hofmeister series (Equation (1)) [44] (Figure 4)., i.e., Na_2_SO_4_, KCl, NaCl, NaNO_3_, MgSO_4_, and MgCl_2_. The salts were combined with 8% PEG_6000_ and were applied, according to the previous findings, in an SXC process with ORFV using 3–5 µm membranes and in-line mixing. To generate a standard procedure for loading, and to facilitate a comparison of the results, the maximum loading volume was determined for the 3–5 µm membranes using “standard” conditions. Here, the feed consisted of a 20 mM TRIS-HCl buffer with 180 mM NaCl, and the elution of the same TRIS buffer, but with 400 mM NaCl. The pressure limit of 0.4 MPa was reached after a 40 mL load. This procedure was adapted to ensure the use of the membrane modules’ full binding capacity, thus reducing the relative loss due to unspecific binding.

In contrast to the previous experiments, an additional second elution step after the standard elution in the direction of flow, down-flow (DF), was included in this set-up, using countercurrent flow (up-flow, UF). Comparing the impurity removal of protein (BCA protein assay, Thermo Fisher Scientific) and dsDNA (Quant-iT PicoGreen dsDNA assay, Thermo Fisher Scientific), no differences were detected for the application of the different salts, i.e., less than 1% residual protein and 25–40% residual dsDNA were found in the elution fraction (data not shown).
(1)$$\overset{\mathrm{chaotropic}}{\to }\;{\mathrm{SO}}_{4}^{2-}<{\mathrm{HPO}}_{4}^{2-}<\mathrm{CH}3{\mathrm{COO}}^{-}<{\mathrm{Cl}}_{}^{-}<{\mathrm{NO}}_{3}^{-}<{\mathrm{CLO}}_{3}^{-}<I_{}^{-}\;{\mathrm{NH}}_{4}^{+}<K_{}^{+}<{\mathrm{Na}}_{}^{+}<{\mathrm{Mg}}_{}^{2+}<{\mathrm{Ca}}_{}^{2+}\;\underset{\mathrm{kosmotropic}}{\leftarrow }$$

#### 3.4.1. Influence of Salts on SXC According to the Hofmeister Series

First, the different salts, i.e., Na_2_SO_4_, KCl, NaCl, NaNO_3_, MgSO_4_, and MgCl_2,_ were applied at a 20 mM final concentration in the SXC and in the precipitation studies. The precipitation study revealed two pronounced effects: Compared to a negative control without salt addition (2500 nm), both salts containing 20 mM Mg^2+^ increased the ORFV aggregate size up to 4000 nm after 2 h, whereas Na_2_SO_4_ reduced the aggregate size to 1500 nm (Figure 5B). Further, NaCl, KCl, and NaNO_3_ had no impact on the precipitation at a 20 mM concentration. The size of the aggregates correlated with the SXC yields (Figure 5A). The highest ORFV yields were found for 20 mM MgCl_2_ (84%), followed by MgSO_4_ (65%). The other four salts, Na_2_SO_4_, KCl, NaCl, and NaNO_3_, indicated no significant differences in the individual SXC fractions. However, the addition of 20 mM Na_2_SO_4_ yielded roughly 10% more infectious ORFV in the elution fraction.

In the presented SXC experimental set-up, PEG is the driving force for precipitation and, after water, the most abundant solute [9]. Thus, changes in the precipitation efficiency should be dominated by the influence of salts on the PEG. In several preceding studies, different salts were applied in the SXC, which were described to mainly cause the following effects, some of them well-known from precipitation [9,10,11,23] (Figure 4). The first of these effects is the reduction of the target retention at high salt concentrations (approx. > 0.3 M), caused by a compacting of the PEG molecules [9,10,11]. Hence, the size of the PEG-deficient zone is reduced, suppressing the exclusion effect. This effect has been highlighted with salts following the Hofmeister series, meaning more kosmotropic salts show a stronger negative influence on the retention, as shown for the impact of (NH_4_)_2_SO_4_ as compared to NaCl on an immunoglobulin G retention by Lee et al. [9]. Secondly, at low salt concentrations (<0.2 M), the pH is the dominant variable, as it determines the charge (repulsion) of the targets, which is more pronounced for targets with a high charge density, such as phages or nucleic acids [9,10]. By increasing the salt concentration, the process can benefit from charge shielding, which facilitates a higher density packing of the targets [23]. Last, specific ion-binding can alter the charge distribution of the surface proteins of the viral particles, as was described for phosphates [45].

The observation of an increased yield with Na_2_SO_4_ in the SXC process was unexpected. Theoretically, this kosmotropic salt should have had a negative effect on the yield, as described before [9], which was supported by the precipitation kinetics, showing reduced efficiencies (Figure 5B,C). We propose that the 10% increase in target retention might be due to the ability of strongly kosmotropic salts to retain targets on uncharged surfaces, as was summarized by Arakawa and Gagnon [46] for size exclusion and ion exchange media. It remains to test higher concentrations of Na_2_SO_4_ to evaluate the effect of the salt-induced target retention on the stationary phase. Such tests could also be insightful for further investigations on the solubility maximum of the ORFV in the presence of kosmotropic salts. The pre-column pressure and loading volumes for the different salts were comparable (Appendix A). With the here tested range (20–200 mM) for the ORFV precipitation, we could not make a distinction between the impact of PEG-compacting and the target solubility induced by the salts (Figure 4).

#### 3.4.2. Concentration-Dependent Precipitation

Next, the precipitation kinetics of the aforementioned salts, Na_2_SO_4_, KCl, NaCl, NaNO_3_, MgSO_4_, and MgCl_2,_ combined with ORFV, was analyzed up to 200 mM. Concerning the Mg^2+^ salts, the increase in the precipitation efficiency remained constant for tests with 20, 50, and 100 mM MgCl_2_ (~4000 nm), but decreased, as observed before, for 200 mM (3000 nm) (Figure 5C). A similar effect was visible for NaNO_3_, but with smaller sizes of 1000–1500 nm. MgSO_4_, on the contrary, revealed a complete prevention of precipitation at 100 and 200 mM. Na_2_SO_4_ followed the same trend as MgSO_4_. The other two salts, NaCl and KCl, showed no concentration-dependent impact in the precipitation behavior of the cell culture-derived ORFV.

This picture allows for several salt-specific assumptions. Firstly, PEG-compacting is the dominant variable at the maximum salt concentration (200 mM) for MgCl_2_ and NaNO_3_. Secondly, with increasing concentrations, kosmotropic ions (SO_4_^2-^) control the precipitation of the ORFV, instead of the chaotropic ions (Mg^2+^). Thus, the influence of divalent ions might be stronger, compacting the PEG at a smaller concentration. Divalent ions are also known to be more effective charge-shielding agents, facilitating the precipitation behavior of proteins (Appendix A). Thus, thirdly, this could be the cause for the increase in the precipitation efficiency for MgCl_2_ and NaNO_3_, which was not observed for NaCl and KCl. Fourthly, the protein solubility does not dictate the precipitation process. Mg^2+^ was reported to bind to protein surfaces [47], increasing the solubility of viruses [48]. Counteracting this hydration, Mg^2+^ is categorized as more chaotropic, which is known to increase protein solubility (salting-in) up to a maximum plateau [49], preventing the crowding-out (Figure 4). Kosmotropic compounds, on the contrary, themselves undergo a preferential exclusion in a salt-protein system [46,47]. They induce a preferential hydration of the virus surface [47], enhancing the precipitation in a concentration-dependent manner, with a minimum at low molarities (<0.5 M) [49]. Clearly, these solubility effects of the proteins are of an opposite nature for the presented results. We assume that the effects on the PEG are dominant in this system, and the concentrations of the salts were too low to show solubility-dependent effects.

#### 3.4.3. Concentration-Dependent Impact of NaCl and NaNO_3_

In Figure 5C, three distinct curve progressions were observed for the salt concentration-dependent PEG precipitation of the ORFV: (1) NaCl and KCl did not influence the aggregate size, as compared to the control without a salt addition; (2) NaNO_3_ and MgCl_2_ revealed curves with a size maximum between 100 and 200 mM; and (3) the two SO_4_^2-^ salts showed a maximum at the lowest tested concentration (20 mM) with an asymptotic decrease.

Here, we selected NaCl and NaNO_3_ from the group (1) and (2), respectively, to investigate the correlation between the ORFV cell culture precipitate size and the SXC recovery. Just as for the experiments with 20 mM salt, the ORFV retention in the SXC correlated with the aggregate size for NaNO_3_, but not for NaCl. In detail, for NaNO_3,_ the aggregate size was maximal with 1900 nm at 50 and 100 mM (Figure 5C). The ORFV yield in the SXC processes suggested a similar behavior (Figure 6A) with a maximum of 64% ORFV yield at 100 mM NaNO_3_. Interestingly, the differences in the aggregate size did not correlate in a quantitative manner with the SXC yields. This was even more pronounced for NaCl. Here, the salt concentration caused no differences in the aggregate size in the range of 0 to 200 mM (Figure 5C). The yield in the SXC process, however, was elevated with increasing concentrations from 26%, to 51%, to 61% for 20, 50, and 200 mM (Figure 6B). The increase in yield was comparative to the findings mentioned by Lee et al. [9] and Levanova and Poranen [23], where reduced mutual charge repulsion led to a higher density packing of the targets. The first study predicted the concentration necessary for a sufficient charge-shielding at 200 mM NaCl for phage M13K07. In our study, the highest yield for NaCl was predicted to be in the range of at least 200 mM.

Observing this complex behavior, precipitation kinetics, as conducted in this research, can indicate increasing or decreasing SXC efficiencies, however, not for every case. More research is necessary to understand the complex interactions within the chromatographic columns, particularly, if additional salts are applied.

## 4. Conclusions

In this study, we have summarized practical considerations for the application and characterization of the SXC. We focused especially on three factors with a considerable influence on the SXC performance for an ORFV purification, which have not yet been extensively characterized before: (1) the comparison of pore sizes of the stationary phases, (2) the contact time of PEG and the target, and (3) a variety of salts and their concentrations. While smaller pores of the stationary phase reduce the necessary PEG concentrations, bigger pores induce a lower backpressure with similar yields. Especially, if extended contact times of PEG and the target are expected, increased precipitate sizes might cause unwanted filtration effects when using small-pored membranes. Further care should be taken in the selection of buffers and other salts, due to their influence on the precipitation process. Yield losses were encountered when applying kosmotropic substances, such as citrate. However, Mg^2+^, strongly chaotropic, acted beneficial on the SXC process. To further understand these interactions, we propose that the analysis of the surface protein composition of the target is beneficial for predicting salt-specific effects on the preferential depletion mechanism. Last, although a quantitative prediction was not possible, the precipitation success correlated qualitatively in most cases. Thus, we conclude that offline precipitation kinetics can be a valuable tool to predict the target retention in SXC. In summary, we suggest the use of cellulose membranes with 3–5 µm pore diameter for a stable ORFV purification. Although the impurity removal was not affected, the load volume could be increased with these membranes. Additionally, we expect Mg^2+^ salts to increase the virus binding to the membrane column in SXC without increasing the PEG concentration.

## Figures and Tables

**Figure 1 membranes-12-01070-f001:**
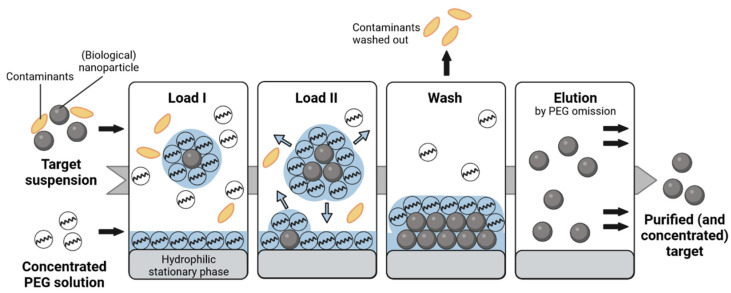
Overview of the working principle of the steric exclusion chromatography (SXC). The target (including impurities from production) is mixed with a concentrated PEG solution. The feed is loaded onto a hydrophilic stationary phase, e.g., cellulose membranes. A PEG-deficient zone (blue) covers the molecules and the stationary phase (load I), while the thickness of this zone is determined by the hydrodynamic diameter and the concentration of the PEG molecules. To reduce the PEG-deficient zones, targets associate/precipitate with each other as well as at the stationary phase (load II) [9,10]. The binding of the targets at the stationary phase allows for a wash-out of smaller contaminants (wash). The binding is reversable by reducing the PEG concentration (elution) so that a purified, concentrated sample can be eluted. The figure was adapted from Eilts et al. [20], and was prepared using biorender.com.

**Figure 2 membranes-12-01070-f002:**
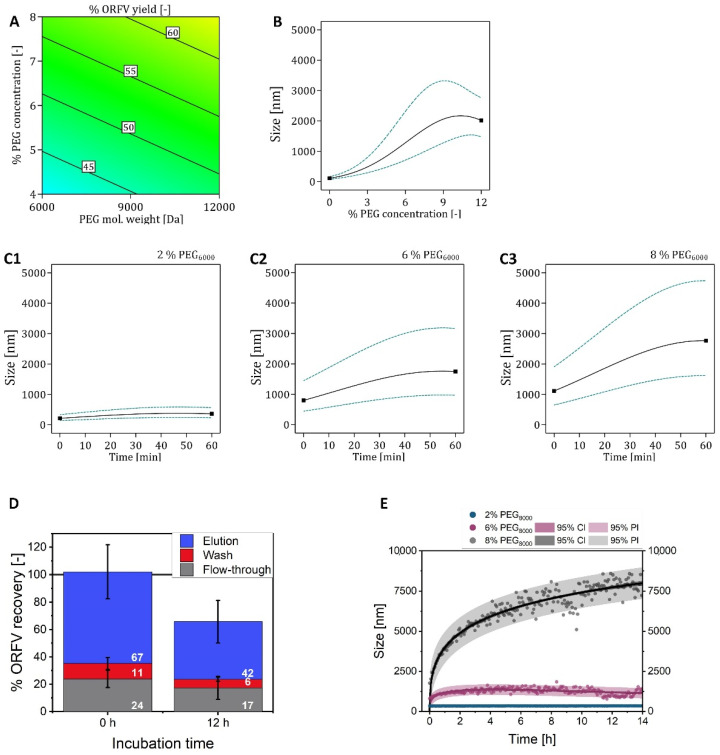
Investigation of loading parameters for steric exclusion chromatography (SXC) of the Orf virus (ORFV). (**A**) The influence of the polyethylene glycol (PEG) molecular weight (6000–12,000 Da) and concentration (4–8%) in a TRIS buffer on the yield of the ORFV was tested, using a design of experiments approach (Appendix A). The flow rate was set to 2 mL min^−1^ and the elution buffer was composed of 0% PEG + 0.4 M NaCl. (**B**,**C**). According to the conditions chosen in A, the ORFV cell culture supernatant was mixed with concentrated PEG solutions to test the precipitation behavior by size measurements. Samples were incubated in 0–12% PEG_6000_ for 60 min. The resulting kinetics were used to generate a design of experiments-based model (Appendix A). (**C1**–**3**) represent the kinetics for selected concentrations of 2%, 6%, and 8% PEG_6000_. (**D**) Using 8% PEG_8000_ in a citrate phosphate buffer, the SXC of the ORFV was operated using different times of contact of the ORFV and the concentrated PEG solutions. The samples were pre-mixed in a superloop and thereafter either loaded onto the column directly (0 h) or after 12 h of incubation time. Representative chromatograms may be found in the Appendix A. (**E**) Additionally, the same concentrations as in **C1**–**C3**, 2%, 6%, and 8%, were applied with PEG_8000_ in a citrate phosphate buffer in a similar set-up, recording the precipitate size of the ORFV. The reference for the calculations of the individual recoveries was the loading sample after the respective incubation time.

**Figure 3 membranes-12-01070-f003:**
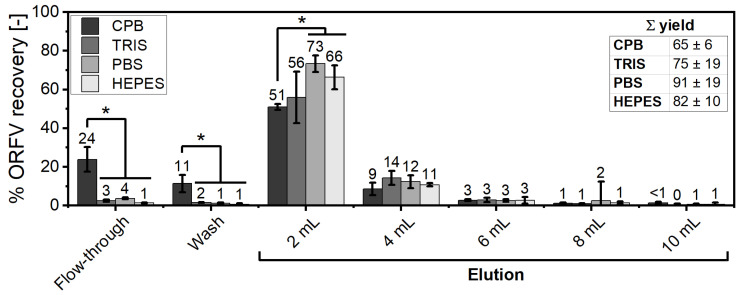
Performance of the steric exclusion chromatography (SXC) depending on the buffer. The Orf virus (ORFV) was processed via SXC (load: 8% PEG_8000_ (polyethylene glycol, 8000 Da); elution: 0% PEG, 0.4 M NaCl) in different buffering systems of neutral pH and physiological conductivity: 0.1 M CPB (dark grey), 20 mM TRIS-HCl (medium grey), PBS (light grey), and 0.1 M HEPES (white). The pre-mixed ORFV/PEG solution was directly applied to the column without an incubation time. The triplicate runs were fractionated into flow-through, i.e., sample application, wash, and elution. The elution itself was fractionated into 5 × 2 mL. Each set of three runs was analyzed regarding the concentration of infectious ORFV. On a volumetric basis, the recoveries were calculated relative to the respective feed. The statistical analysis of the recoveries was performed via ANOVA with a *Tukey* test (*α* = 0.05) for *n* = 3. Asterisks (*) indicate significant differences within the respective group. Representative chromatograms may be found in the Appendix A.

**Figure 4 membranes-12-01070-f004:**
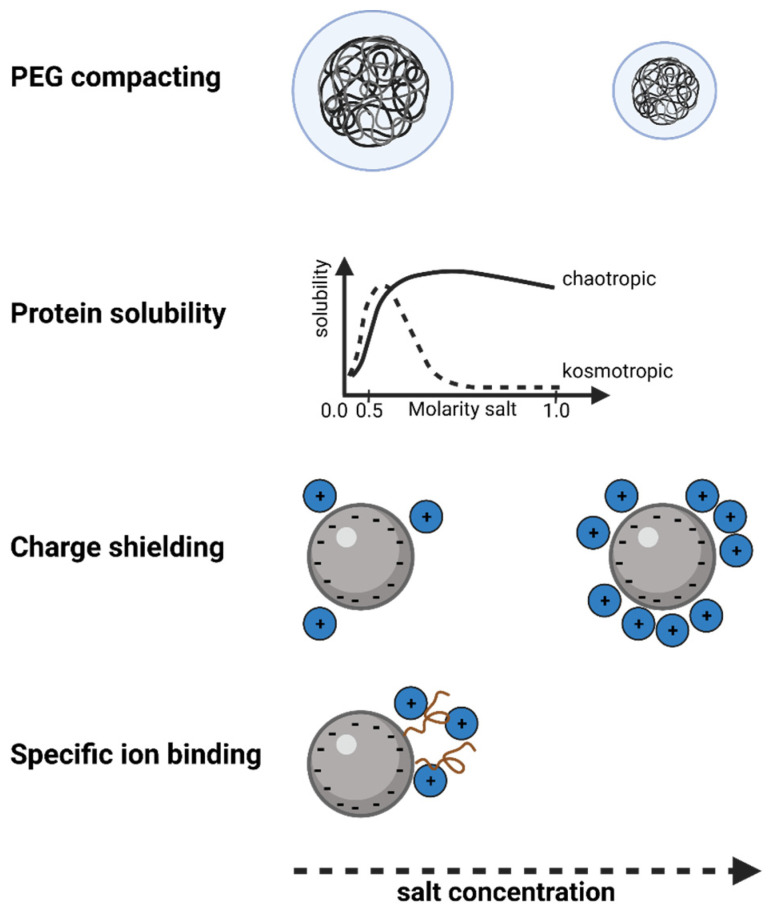
Salt concentration-dependent impact on PEG/salt and salt/protein interactions. Figure prepared using biorender.com.

**Figure 5 membranes-12-01070-f005:**
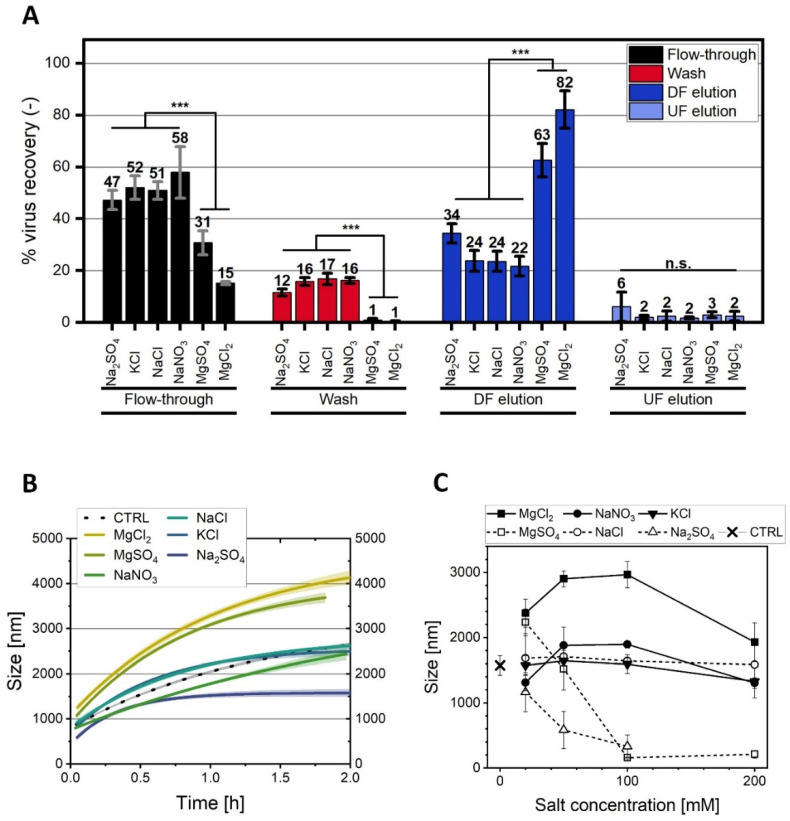
Variation of the salt type throughout the steric exclusion chromatography (SXC). The Orf virus (ORFV) was processed via SXC (load: 8% PEG_6000_ (polyethylene glycol, 6000 Da); elution: 0% PEG, 0.4 M NaCl) in 20 mM TRIS buffer, pH 7.4, supplemented with 20 mM of different salts, i.e., Na_2_SO_4_, KCl, NaCl, NaNO_3_, MgSO_4_, and MgCl_2_. The load was stopped when the pressure limit of 0.4 MPa was reached or 40 mL PEG/ORFV suspension were loaded. The fractions load (flow-through), wash, and the two elution types, down-flow, DF, (in the direction of loading), and up-flow, UF, (countercurrent) were collected and analyzed for the relative ORFV recovery (**A**). (**B**) depicts the time-dependent precipitation kinetics of the cell culture-derived ORFV supernatant with the same conditions as the SXC loads. Using the same salts at 20–200 mM concentration, (**C**) shows the ORFV precipitate sizes after 0.5 h incubation time. CTRL indicates 8% PEG_6000_ without salt addition. The statistical analysis of the recoveries (**A**) was performed via ANOVA with a *Tukey* test (*α* = 0.001) for *n* = 3. Asterisks (***) indicate significant differences within the respective group.

**Figure 6 membranes-12-01070-f006:**
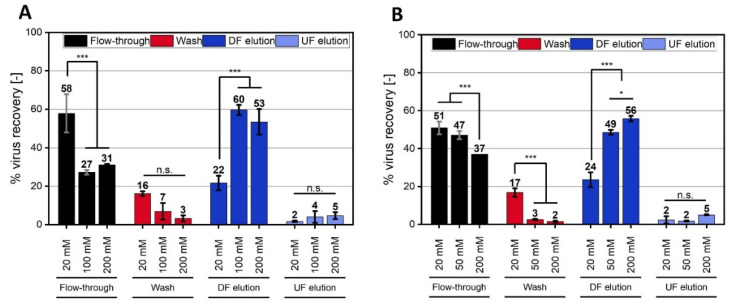
Steric exclusion chromatography (SXC) of the Orf virus (ORFV) with NaCl and NaNO_3_ at concentrations up to 200 mM. The SXC of the ORFV was performed with 8% PEG_6000_ (polyethylene glycol, 6000 Da) in the loading buffer, 0% PEG, and 0.4 M NaCl in the elution buffer. 20 mM TRIS buffer and pH 7.4 were used for all experiments. The load and wash were supplemented with 20–200 mM of NaNO_3_ (**A**) or NaCl (**B**). The load was stopped when the pressure limit of 0.4 MPa was reached, or when 40 mL PEG/ORFV suspension were applied. The runs were fractionated into flow-through (load), wash, and the two elutions, up-flow, UF, (in the direction of loading), and down-flow, DF, (counter-flow). Each fraction was analyzed for its relative ORFV recovery. The statistical analysis of the recoveries was performed via ANOVA with a *Tukey* test (*α* = 0.001) for *n* = 3. Asterisks (***) indicate significant differences within the respective group.

**Table 1 membranes-12-01070-t001:** Overview of recoveries and load volumes for infective ORFV from SXC experiments with regenerated cellulose membranes as stationary phases. Both column types (1 µm and 3–5 µm pore diameter) were single-use. They consisted of ten stacked membranes, each 13 mm in diameter. Each run consisted of a loading step (the flow-through is analyzed), a wash step, and two elution steps, i.e., a down-flow and an up-flow elution. The down-flow elution was performed in direction of flow, while the up-flow elution was counter-current.

Pore Size	Recoveries					Load	
	Flow-Through	Wash	Down-Flow Elution	Up-Flow Elution	Total	Volume	ORFV Particles
[µm]	[%]					[mL]	[IU]
1	3 ± 1	3 ± 1	99 ± 17	21 ± 10	123 ± 29	8 ± 5	1.8 × 10^7^ ± 1.1 × 10^7^
3–5	37 ± 1	2 ± 1	56 ± 1	5 ± 1	100 ± 2	31 ± 1	7.0 × 10^8^ ± 2.2 × 10^6^

## Data Availability

Not applicable.

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
