# Peer review of "A Summary of Practical Considerations for the Application of the Steric Exclusion Chromatography for the Purification of the Orf Viral Vector"

_membranes, 2022, doi:10.3390/membranes12111070_

Round 1

Reviewer 1 Report

An upfront clear explanation of SXC using graphics is required. This is not a very mainstream technique and most reader of Membranes are likely to be unaware of it.

Even thought membrane are used in the separation, they are not used in the sense they are used in membrane science. This paper is better suited for a chromatography journal. 

The language is quite nebulous at times. Sentence structure could be significantly improved,

Information and graphics on equipment used is scarce or entirely missing. Once again, this is not a very mainstream technique.

Experimental protocols should be more clearly stated.

It is surprising to see a chromatography paper with no chromatograms. 

The figures could be improved quite significantly.

Most importantly, the main strategy involved in separation needs to be clearly stated. The method seems to be like a blackbox with a bunch of inputs and outputs.

Overall, there is significant room for improvement.

Author Response

Response to Reviewer 1 Comments

We would like to thank the reviewer for their efforts and valuable suggestions to improve the manuscript, which we have taken into account in the resubmitted version.

Point 1: An upfront clear explanation of SXC using graphics is required. This is not a very mainstream technique and most reader of Membranes are likely to be unaware of it.

Response 1: We appreciate this comment and improved the explanation of the SXC including Figure 1 in the introduction.

Point 2: Even thought membrane are used in the separation, they are not used in the sense they are used in membrane science. This paper is better suited for a chromatography journal. 

Response 2: Thank you for pointing towards to scope of the journal. Membranes are commonly applied in SXC, and, in this research, the use of different cellulose membrane types was evaluated. We therefore agree with the reviewer that the manuscript would be suitable for a chromatography journal, however, we want to point out that it is a suitable follow-up paper to a recent SXC publication in membranes (doi: 10.3390/membranes12050488).

Point 3: The language is quite nebulous at times. Sentence structure could be significantly improved,

Response 3: The manuscript was reviewed and improved in the respective sections. Additionally, spell checking was performed by a native English speaker.

Point 4: Information and graphics on equipment used is scarce or entirely missing. Once again, this is not a very mainstream technique.

Response 4: Thank you for this comment. We added more detailed descriptions to the materials and methods section, improving the description of the experimental protocols and the equipment used.

Point 5: Experimental protocols should be more clearly stated.

Response 5: Please refer to the answer to point 4.

Point 6: It is surprising to see a chromatography paper with no chromatograms. 

Response 6: We agree that chromatograms are a valuable tool to understand the impact of process parameters on chromatographic techniques, and the respective chromatograms can be found in the supplementary material. We hope the reviewer appreciates that in this manuscript, the important statements are explained by the provided figures in the main manuscript.

Point 7: The figures could be improved quite significantly.

Response 7: We appreciate the comment on figure quality and improved the resolution.

Point 8: Most importantly, the main strategy involved in separation needs to be clearly stated. The method seems to be like a blackbox with a bunch of inputs and outputs.

Response 8: This point is related to point 1. Please refer to the answer to point 1.

Reviewer 2 Report

The manuscripts describes a series of experiments that have been performed to enhance the understanding on the purification of ORFV. However, the current work needs some rewriting and representing some additional data to make the work more comprehensive to scientist that are not familiar with the SXC or work with viral vectors.

A general comment: Does this manuscript fill the scope of the journal Membranes: “Membranes provides a forum for publishing papers that advance our understanding of membrane structure, performance, processes, and applications that cover membrane chemistry, physics, engineering, and biology”. The introduction part does not link with the scope of the Membranes and this could be improved.

Another general comment: I feel that the text would be much easier to follow if the text on lines 56-75 would be supported with a figure to illustrate how the exclusion actually operates.

Another general concern: Comparison of the number of infectious viruses in the control solvent as well as in different salt and PEG concentrations and Mws used is important for the interpretation of the results. It appears that this is done but not cited in the main text (Fig. S1.).

It was also stated that the amount of the loaded viruses was either a certain volume or restricted by the pressure limits/backpressure. I would like to see the data on the different loading amounts that the different buffers and column combinations “accepted “to be able to compare the different applications.

The authors should check that the data in the figures is cited in the text. This includes the supplementary material.

Chromatography is not only about the recovery but also about the purification. So please represent data on how the purity of the sample improved. One should show how the protein patterns of the loaded sample and purified sample changed, was there a change in the specific infectivity (virus number/mg per protein)? Was some contaminating nucleic acids removed that most probably are present in the virus lysates.

How much was the virus recovered from the purification diluted, also an important aspect affecting the success of chromatography.

Minor points:

Please describe in more details the advantages of SXC lines 49-55. I would assume that PEG precipitation is much easier to scale up than chromatography so what is meant by that the scale up is simplified?

Line 45: By this variety of influencing factors… please rewrite

Line 48: Please explain how chromatography allows higher throughput compared to precipitation.

Line 56: In most applications, PEG is utilized to ….

Lines 76-81. I feel that for the reader to be able to understand the challenges with this ORF vector, it should be better described (what does it constitute of, what is its size (Rh)). Start the chapter with introduction of the vector and then describe the previous studies on its purification. Also combine the work that was done in this manuscript to a single chapter lines 76-93)

Line 85: I would remove the” i.e. the ORFV cell culture supernatant combined with PEG” as I feel it is confusing and unclear.

Lines 118-119: please give the pH of the used buffers

Size measurements: Please indicate the DLS name and manufacturer that was used for Rh measurements: in Mat met section.

Line 130 maybe delete with varying systematic parameters, this is evident from the following sentences.

Line 131: The cell culture supernatant was mixed with…

Lines 134-135: Maybe remove the information on the buffers and include the data in the following sentences. 20 mM Tris-HCl (pH 7.4) was utilized to for experiments testing varying salt conditions (20-200 mM). CPB was used for screening …

design space, not evident to me what it means

161-163: Maybe the incubation time for PEG precipitation could be mentioned as it eases up the comparison to 3.1.2

165-166: with and optimum at the edge of design space at … this could be clarified.

166-167: The slopes are quite similar looking, maybe numerical values could be given if the difference in the slope really is significant.

172: global maximum? previously reported maximum?

173-175: Please rephrase. Higher PEG concentrations were not tested as the high viscosity hampers processing and prevents application of high flow rates during chromatography. So does this impact the scale up process that was said to be easy in upper section of the manuscript.

Lines 226-230: Please clarify/rewrite the following sentences as it was difficult to understand. Include a reference for the assumption that accumulations of smaller precipitates might be more sensitive to a shear stress-induced break-up. I did not follow the idea between aggregate dissociation and pre-column pressure.

line 241: Without incubation, the total ORFV recovery was 102 ± 30 % while an incubation time of 12 h revealed 66 ± 20 % (Figure 1D). Thus, a mean of 34 % was not recovered from the column for the runs after 12 h incubation time. Accordingly, the ORFV yield for no incubation time showed 67 ± 20 %, and 42 ± 8 % for 12 h. It is not clear how the recovery and yield differ, please clarify. Are the first number prior to chromatography, but after precipitation (if performed) and the second after the actual chromatography event?

Line 256: and such effects were not observed in previous experiments with the ORFV (the reference is missing).

Line 277: the yields remained high with 99 % ± 17 % and full total recoveries (Table 1). If the yield is 99%, isn’t that full recovery? Was the experiment performed with what PEG concentration and what buffer? Indicate this in mat met section or in the text of section 3.2.

Line 300: A considerable fraction of the ORFV was eluted from the 1 μm membranes (20 %), whereas the yield from bigger pores was increased by only 5 %.

Figure 2: Was the number of infectious viruses affected in the load by the used buffers? Is the recovery based on the number of infectious viruses in the load in the corresponding buffer.

Figure 1: Panels should be presented in the order of A,B, C, D E in the legend.

line 347: It can be concluded here that the choice of the buffering system has a significant influence on the SXC performance, and should be taken with care. Did you observe changes in the infectivity of viruses with the used buffers?

Fig. S5 shows very different UV signals for the eluent thus suggesting very different concentration of purified virus or contaminants. Were the fractions analyzed for the presence of contaminants? What explains the difference in UV but not in virus recovery that was 65 to 91%. The UV peak is much broader than the LS signal in Hepes buffer. LS measurement has increased sensitive for larger particles, suggesting that the tail visible with the UV contains impurities? Same phenomena was seen in Fig. S6.

Supplementary material is extensive.

Conclusions could give the reader some combinations of buffer, column, salt, PEG Mw that based on the research 1) give the best purity of the virus, 2) gives the best recovery of infectious viruses, 3) allows the highest load of viruses; 4) gives the highest concentration of virus particles (number of particles/ml).

Author Response

Response to Reviewer 2 Comments

We would like to thank the reviewer for their efforts and valuable suggestions to improve the manuscript, which we have taken into account in the resubmitted version.

Point 1: A general comment: Does this manuscript fill the scope of the journal Membranes: “Membranes provides a forum for publishing papers that advance our understanding of membrane structure, performance, processes, and applications that cover membrane chemistry, physics, engineering, and biology”. The introduction part does not link with the scope of the Membranes and this could be improved.

Response 1: Thank you for pointing towards to scope of the journal. Membranes are commonly applied in SXC, and in this research the use of different cellulose membrane types was evaluated. We improved the introduction to point towards the scope of the journal Membranes more clearly.

Point 2: Another general comment: I feel that the text would be much easier to follow if the text on lines 56-75 would be supported with a figure to illustrate how the exclusion actually operates.

Response 2: We appreciate this comment and improved the explanation of the SXC including a figure (Figure 1) in the introduction.

Point 3: Another general concern: Comparison of the number of infectious viruses in the control solvent as well as in different salt and PEG concentrations and Mws used is important for the interpretation of the results. It appears that this is done but not cited in the main text (Fig. S1.).

Response 3: Thank you for pointing this out. We included the following information in section 2.3: The presence of PEG did not affect the ORFV infectivity, which was shown before by Lothert et al by complete recovery of the virus even after several freeze-thaw cycles (doi: 10.1016/j.jbiotec.2020.07.023). The effect of salts in the ORFV infectious titer was tested and described in the Supplementary Material S1.

Point 4: It was also stated that the amount of the loaded viruses was either a certain volume or restricted by the pressure limits/backpressure. I would like to see the data on the different loading amounts that the different buffers and column combinations “accepted “to be able to compare the different applications.

Response 4: We agree with the reviewer that this topic needed more explanation. We included a further description for the two membrane types in the materials and methods section 2.2: “the feed was loaded onto the membranes until the pressure limit of the respective membrane module was reached, i.e., 2.0 MPa for 1 µm membranes and 0.4 MPa for 3 - 5 µm.” The results of this comparison are summarized in Table 1. The pressure was chosen as limiting parameter for the SXC experiments if the membrane module was characterized. In the sections 3.1.3, 3.3., and 3.4, the loading volume was fixed to ease the comparison of the experiments. We added this information in the respective sections. In the sections 3.1.2 and 3.3, 10 mL feed was used as loading volume with the intention to compare the two membrane modules. In section 3.4, the comparison of the different salts was the aim. Thus, with the 3-5 µm modules, the loading volume under standard conditions (20 mM TRIS-HCl + 180 mM NaCl for loading and the same TRIS buffer with 0.4 M NaCl for elution (Lothert at al., doi: 10.1016/j.jbiotec.2020.07.023)) was investigated and used for follow-up runs, to standardize the procedure. We did not test the maximum loading capacity for the sections 3.3 and 3.4, but rather observed the pressure signal (see Supplementary Material S4.3).

Point 5: The authors should check that the data in the figures is cited in the text. This includes the supplementary material.

Response 5: Thank you for this comment. We improved the manuscript concerning the references of all figures.

Point 6: Chromatography is not only about the recovery but also about the purification. So please represent data on how the purity of the sample improved. One should show how the protein patterns of the loaded sample and purified sample changed, was there a change in the specific infectivity (virus number/mg per protein)? Was some contaminating nucleic acids removed that most probably are present in the virus lysates.

Response 6: We appreciate the comment and added more details on sample purity concerning residual protein (< 1 %) and dsDNA (25 – 40 %) after SXC application in section 3.4, which was considerably constant. However, in this research, we focused on understanding changes in the virus recovery rather than impurity removal.

Point 7: How much was the recovered virus from the purification diluted, also an important aspect affecting the success of chromatography.

Response 7: We agree with the reviewer that concentration is a relevant topic. Like the impurity removal, this was not the focus of this research. Nevertheless, we discussed this in section 3.3., where different elution volumes are investigated. By using 10 mL load and 10 mL elution, as was done for most experiments, the virus is diluted 1.25 x. however, in section 3.3, we investigated that 4 mL elution volume are enough for this specific application, loosing less than 5 % virus. Thus, the applied ORFV was concentrated nearly 2 x.

Point 8: Please describe in more details the advantages of SXC lines 49-55. I would assume that PEG precipitation is much easier to scale up than chromatography so what is meant by that the scale up is simplified?

Response 8: We restructured the paragraph, including the need for a secondary step after precipitation, which might be omitted using SXC. This was meant by “scale-up”, which might have been misleading.

Point 9: Line 45: By this variety of influencing factors… please rewrite

Response 9: This sentence was restructured to clarify the influencing factors. In lines 45ff.

Point 10: Line 48: Please explain how chromatography allows higher throughput compared to precipitation.

Response 10: Please refer to the answer to point 8.

Point 11: Line 56: In most applications, PEG is utilized to ….

Response 11: This sentence was rewritten in lines 72f.

Point 12: Lines 76-81. I feel that for the reader to be able to understand the challenges with this ORF vector, it should be better described (what does it constitute of, what is its size (Rh)). Start the chapter with introduction of the vector and then describe the previous studies on its purification. Also combine the work that was done in this manuscript to a single chapter lines 76-93)

Response 12: We agree with the reviewer and restructured the whole paragraph in lines 103-124.

Point 13: Line 85: I would remove the” i.e. the ORFV cell culture supernatant combined with PEG” as I feel it is confusing and unclear.

Response 13: Thank you for the suggestion, which we included in the new version of the manuscript.

Point 14: Lines 118-119: please give the pH of the used buffers

Response 14: The pH of all buffers used in SXC was pH 7.4. We clarified this detail in the materials and methods section 2.2.

Point 15: Size measurements: Please indicate the DLS name and manufacturer that was used for Rh measurements: in Mat met section.

Response 15: Thank you for pointing this out. We added the information in the respective section 2.4: “Zetasizer ZS90 with the respective Zeatsizer software, version 7.13, (both Malvern Panalytical) in micro-cuvettes (Sarstedt)”

Point 16: Line 130 maybe delete with varying systematic parameters, this is evident from the following sentences.

Response 16: We appreciate this comment and removed the mentioned text passage.

Point 17: Line 131: The cell culture supernatant was mixed with…

Response 17: We omitted “therefore” in this sentence (lines 172.).

Point 18: Lines 134-135: Maybe remove the information on the buffers and include the data in the following sentences. 20 mM Tris-HCl (pH 7.4) was utilized to for experiments testing varying salt conditions (20-200 mM). CPB was used for screening …

Response 18: Thank you for this suggestion, which we implemented in lines 173ff.

Point 19: design space, not evident to me what it means

Response 19: Thank you for raising this topic. We included more information on the design of experiment approach used and the term “design space” in section 2.3 and where the results of the DOEs are presented.

Point 20: 161-163: Maybe the incubation time for PEG precipitation could be mentioned as it eases up the comparison to 3.1.2

Response 20: We agree with this point and included the information in lines 208f.

Point 21: 165-166: with and optimum at the edge of design space at … this could be clarified.

Response 21: Please see the answer to point 19.

Point 22: 166-167: The slopes are quite similar looking, maybe numerical values could be given if the difference in the slope really is significant.

Response 22: To clarify this point, more information on the impact of the two parameters PEG concentration and MW was added in lines 214ff.

Point 23: 172: global maximum? previously reported maximum?

Response 23: We agree with the reviewer that the term might be misleading. With global maximum we mean the maximum that could be reached in a wider range of the parameters. We changed the term to “optimum” in line 222.

Point 24: 173-175: Please rephrase. Higher PEG concentrations were not tested as the high viscosity hampers processing and prevents application of high flow rates during chromatography. So does this impact the scale up process that was said to be easy in upper section of the manuscript.

Response 24: Thank you for this detailed observation. We included the limitations of high viscosities in SXC in lines 223ff. and in the introduction in lines 97ff.

Point 25: Lines 226-230: Please clarify/rewrite the following sentences as it was difficult to understand. Include a reference for the assumption that accumulations of smaller precipitates might be more sensitive to a shear stress-induced break-up. I did not follow the idea between aggregate dissociation and pre-column pressure.

Response 25: Thank you for the suggestions, which we included in the manuscript with: “The accumulations of smaller precipitates in loose formations might be more sensitive to a shear stress-induced break-up compared to dense big aggregates. Accordingly, we assume that the actual size of the aggregates loaded onto the membranes was smaller than the size measured in the static experiments shown here. Thus, this effect might be the cause that the precipitates of virus and cell debris, several times bigger than the pore size of the column used in the previous section (1 µm), did not increase the pre-column pressures significantly by inducing pore blockage on the top membrane layers.”

Point 26: line 241: Without incubation, the total ORFV recovery was 102 ± 30 % while an incubation time of 12 h revealed 66 ± 20 % (Figure 1D). Thus, a mean of 34 % was not recovered from the column for the runs after 12 h incubation time. Accordingly, the ORFV yield for no incubation time showed 67 ± 20 %, and 42 ± 8 % for 12 h. It is not clear how the recovery and yield differ, please clarify. Are the first number prior to chromatography, but after precipitation (if performed) and the second after the actual chromatography event?

Response 26: Thank you pointing this out. We clarified the difference in lines 301ff. Thank you pointing this out. The term “yield” refers to the recovery in the elution fraction, while recovery might be any fraction. Total recovery are all fractions combined.

Point 27: Line 256: and such effects were not observed in previous experiments with the ORFV (the reference is missing).

Response 27: Thank you for pointing this out. We included the respective references.

Point 28: Line 277: the yields remained high with 99 % ± 17 % and full total recoveries (Table 1). If the yield is 99%, isn’t that full recovery? Was the experiment performed with what PEG concentration and what buffer? Indicate this in mat met section or in the text of section 3.2.

Response 28: We agree that the mentioning of the full recoveries is obsolete. Additionally, we included specifications on the buffers in the materials and methods section.

Point 29: Line 300: A considerable fraction of the ORFV was eluted from the 1 μm membranes (20 %), whereas the yield from bigger pores was increased by only 5 %.

Response 29: Could you please specify what is meant by this point?

Point 30: Figure 2: Was the number of infectious viruses affected in the load by the used buffers? Is the recovery based on the number of infectious viruses in the load in the corresponding buffer.

Response 30: Thank you for the detailed question. The buffers did not affect the virus infectivity in the time frame of SXC experiments, which we described in section 2.3. This research will be published in another manuscript and is therefore not presented here in detail.

Concerning the second question, yes, the recovery is always normalized to the respective feed as described in section 2.2.

Point 31: Figure 1: Panels should be presented in the order of A,B, C, D E in the legend.

Response 31: Thank you for pointing this out. We corrected the mistake in Figure 2.

Point 32: line 347: It can be concluded here that the choice of the buffering system has a significant influence on the SXC performance, and should be taken with care. Did you observe changes in the infectivity of viruses with the used buffers?

Response 32: Please refer to the answer to point 30.

Point 33: Fig. S5 shows very different UV signals for the eluent thus suggesting very different concentration of purified virus or contaminants. Were the fractions analyzed for the presence of contaminants? What explains the difference in UV but not in virus recovery that was 65 to 91%. The UV peak is much broader than the LS signal in Hepes buffer. LS measurement has increased sensitive for larger particles, suggesting that the tail visible with the UV contains impurities? Same phenomena was seen in Fig. S6.

Response 33: We agree with the reviewer that the UV signals in figures S5 and S6 might be misleading concerning the shape compared to the data on protein and virus recovery in the respective elution fractions. We suspect this to be due to aggregates elution from the column. The DLS signal is usually a much better reference for virus recovery in the individual fractions, while UV is a good indication for the comparability of the load.

Point 34: Supplementary material is extensive.

Response 34: We agree with the reviewer on this matter. However, to present all background data, we included detailed descriptions in the supplementary material.

Point 35: Conclusions could give the reader some combinations of buffer, column, salt, PEG Mw that based on the research 1) give the best purity of the virus, 2) gives the best recovery of infectious viruses, 3) allows the highest load of viruses; 4) gives the highest concentration of virus particles (number of particles/ml).

Response 35: Thank you for this suggestion. We included some recommendations in the conclusion section lines 583ff.
